

# The biomarkers discovery of hyperuricemia and gout: proteomics and metabolomics

Xinghong Wu and Chongge You

Laboratory Medicine Center, Lanzhou University Second Hospital, Lanzhou, Gansu, China

## ABSTRACT

**Background.** Hyperuricemia and gout are a group of disorders of purine metabolism. In recent years, the incidence of hyperuricemia and gout has been increasing, which is a severe threat to people's health. Several studies on hyperuricemia and gout in proteomics and metabolomics have been conducted recently. Some literature has identified biomarkers that distinguish asymptomatic hyperuricemia from acute gout or remission of gout. We summarize the physiological processes in which these biomarkers may be involved and their role in disease progression.

**Methodology.** We used professional databases including PubMed, Web of Science to conduct the literature review. This review addresses the current landscape of hyperuricemia and gout biomarkers with a focus on proteomics and metabolomics.

**Results.** Proteomic methods are used to identify differentially expressed proteins to find specific biomarkers. These findings may be suggestive for the diagnosis and treatment of hyperuricemia and gout to explore the disease pathogenesis. The identified biomarkers may be mediators of the link between hyperuricemia, gout and kidney disease, metabolic syndrome, diabetes and hypertriglyceridemia. Metabolomics reveals the main influential pathways through small molecule metabolites, such as amino acid metabolism, lipid metabolism, or other characteristic metabolic pathways. These studies have contributed to the discovery of Chinese medicine. Some traditional Chinese medicine compounds can improve the metabolic disorders of the disease.

**Conclusions.** We suggest some possible relationships of potential biomarkers with inflammatory episodes, complement activation, and metabolic pathways. These biomarkers are able to distinguish between different stages of disease development. However, there are relatively few proteomic as well as metabolomic studies on hyperuricemia and gout, and some experiments are only primary screening tests, which need further in-depth study.

Corresponding author
Chongge You, youchg@lzu.edu.cn

## INTRODUCTION

The incidence of hyperuricemia and gout is on the rise, and those suffering from hyperuricemia may be prone to developing gout as well (*Robinson, 2018*). There is a close relationship between hyperuricemia and metabolic syndrome (Mets) (*Chen et al., 2018*), chronic kidney disease (*Kielstein, Pontremoli & Burnier, 2020*), and cardiovascular disease. Generally, hyperuricemia occurs after the levels of uric acid (UA) rise above 420 $\mu$mol L$^{-1}$ (7 mg dL$^{-1}$) for men and over 350 $\mu$mol L$^{-1}$ (6 mg dL$^{-1}$) for women (*Johnson et al., 2003*).

In addition, hyperuricemia is more prevalent in men than women because female gonadal hormones and anti-androgen treatment can prevent the condition (*Wan et al., 2020*). In contrast, serum uric acid (SUA) levels will increase in postmenopausal women. There is a crucial role for UA in hyperuricemia and gout development. Researchers have found that high SUA levels are associated with cardiovascular disease risk factors, and that, as a risk factor, they can increase cardiovascular death rates (*Chales, 2019*; *Borghi et al., 2018*). An analysis of retrospective data in China showed that SUA levels were associated with rapid glomerular filtration rate declines (*Zhou et al., 2019*). As SUA levels rise, urate crystals form in the renal tubules and interstitial spaces, ultimately causing renal damage (*Sivera, Andres & Dalbeth, 2022*; *Su et al., 2020*). A high purine diet or excessive intake of fructose and alcohol can induce the degradation of purine nucleotides, leading to the production of UA (*Zhang, Jiao & Kong, 2017*). The presence of UA is directly related to inflammation in Monosodium urate monohydrate (MSU) crystal-induced gout. However, hyperuricemia is not the only thing that affects a gout flare or attack. Taking into account the physiological characteristics of MSU crystals, the Gout, Hyperuricemia, and Crystal-Associated Diseases Network (G-CAN) has recently defined gout as "a disease caused by MSU crystals" (*Richette et al., 2017*). For the diagnosis of gout, it is essential to determine whether MSU crystals are present in synovial fluid (SF) or gout tophus under the microscope (*Pascart & Liote, 2019*). MSU crystals can also be detected by advanced imaging methods, such as ultrasound and dual-energy computed tomography (*Dalbeth et al., 2021*). Note that even the absence of MSU crystal deposits does not eliminate gout. Gout may present in the form of an acute inflammatory response (gout flare), chronic gouty arthritis, or subcutaneous tophus. Patients with advanced gout can develop persistent joint inflammation and even structural damage over time (*Petty et al., 2019*).

The use of omics technologies, including genomics, proteomics, metabolomics, and transcriptomics, has significantly contributed to the advancement of medicine and biology in recent years (*Monti et al., 2019*). Marc Wilkins first proposed proteomics, that is, all the proteins expressed by a genome or a cell (*Wilkins et al., 1996*). A mass spectrometry (MS) approach to proteomics is crucial for identifying proteins in various diseases (*Mann, 2003*). There are several MS detection strategies that can enhance the sensitivity of MS analysis, such as tandem mass tag (TMT) labeling and isobaric tag for relative and absolute quantification (iTRAQ). Molecular markers for the diagnosis of hyperuricemia and gout are emerging due to proteomic studies in recent years. Metabolomics has been widely used in various fields, such as plant biology, pharmacology, and medicine. There are two analytical methods, non-targeted and targeted metabolomics, which can be based on nuclear magnetic resonance (NMR) spectroscopy or MS techniques. Unlike targeted metabolomics, untargeted metabolomics studies identify novel substances that have a broad, general characterization, while targeted metabolomics studies specific metabolites that are of interest to researchers. Liquid chromatography-mass spectrometry (LC-MS) has been developed as the primary technology platform for metabolic profiling (*Fraga-Corral et al., 2022*). Modern medicine is eventually moving towards precision medicine and pursuing personalized treatment as a benchmark. Many physiological and biochemical changes in human systems can be revealed using "omics" technology, including metabolite changes

in pathological conditions. A combination of proteomics, metabolomics, and biomarker analysis can provide valuable insights into disease diagnosis, prognosis, and therapeutic targets.

The aim of this review is to summarize the biomarkers of hyperuricemia and gout disease as well as metabolic pathways. Our goal is to understand the current state of research regarding biomarkers and how they influence disease progression. We also summarize some types of metabolic disorders present in hyperuricemia and gout, and the improvement of metabolic disorders by some Chinese medicines.

## SURVEY METHODOLOGY

This review is the result of a systematic literature search on PubMed and Web of Science. It was done to find articles related to the proteomics and metabolomics of hyperuricemia and gout from 2010 to 2022. The search terms used for the article in various combinations included "hyperuricemia and proteomics," "gout and proteomics," "hyperuricemia and metabolomics," "gout and metabolomics," and "gout and inflammation". Meanwhile, we consulted the literature on the relationship between hyperuricemia, gout and diabetes, Mets and kidney disease. The search strategy was used to obtain the titles and abstracts of the relevant studies initially screened, and retrieved the full text. We also reviewed the relevant references in the article to ensure comprehensive coverage and no bias in the article.

### The mechanism of hyperuricemia and gout
#### The production and excretion of uric acid

Overproduction of UA is the result of mutations in the UA synthesis gene and changes in its activity. The pathways leading to increased UA production are as follows (Fig. 1). Furthermore, fructose induces the metabolism of ATP and phosphate in the liver and contributes to the accumulation of adenosine monophosphate (AMP). In the next step, AMP generates inosine monophosphate (IMP) under the action of AMP deaminase (AMPD), resulting in UA production (*King et al., 2018*; *Zou, Zhao & Wang, 2021*). After the formation of UA, about 1/3 is excreted from the gastrointestinal tract, and 2/3 passes through the kidney (*Kielstein, Pontremoli & Burnier, 2020*). The proximal convoluted tubules play a dominant role in the reabsorption and secretion of UA in the renal tubules. Eventually, about 90% of UA is reabsorbed into the blood (*Maiuolo et al., 2016*). In humans, the kidneys cannot convert UA into allantoin due to a lack of uricase. Clinical research studies routinely use recombinant forms of uricase, including Rasburicase and PEG, to lower UA levels (*Braga, Foresto-Neto & Camara, 2020*; *Otani et al., 2020*). Additionally, some renal urate transporters, such as URAT1, GLUT9, OAT1, OAT3, and ABCG2, are of critical significance in the active secretion and reabsorption of urate in proximal convoluted renal tubules (*Alghamdi, Soliman & Nassan, 2020*; *Sun et al., 2021*). It is also important for UA levels in the gastrointestinal tract that ABCG2 regulates urate excretion. In cases of urate transporter gene mutations, the structure and function of the gene change, causing urate to accumulate in the kidney and causing renal impairment. As a result, studies based on urate transporters can be beneficial for disease prevention. For example, PF-06743649

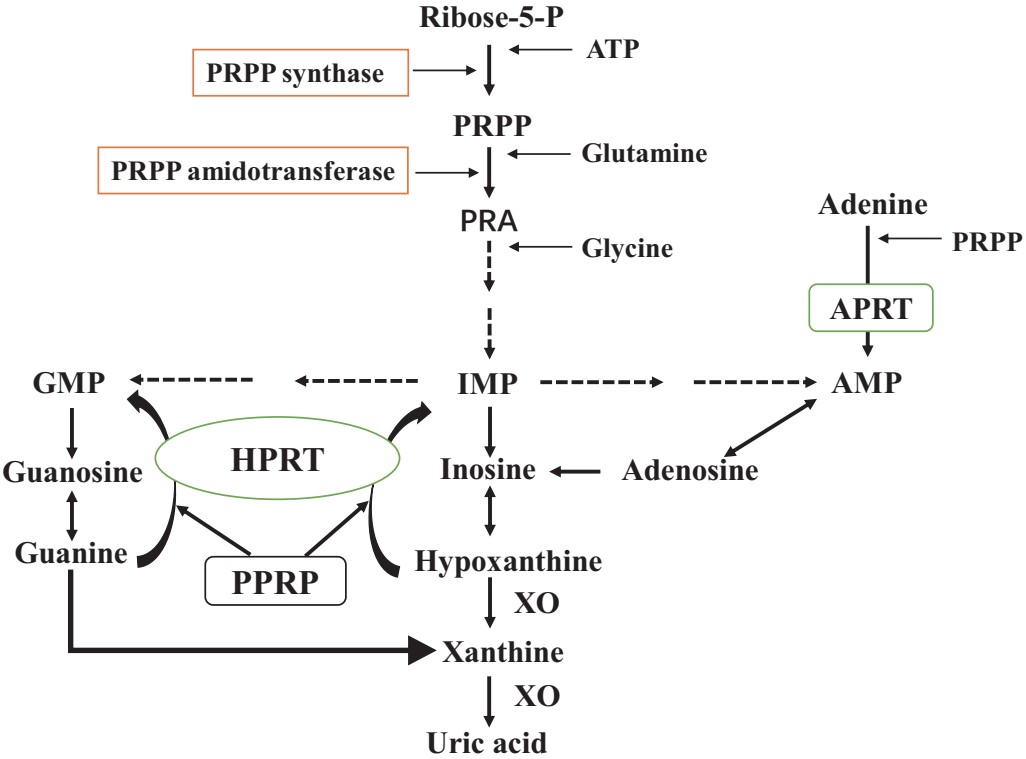

**Figure 1** *De novo* **synthesis and salvage pathway of purine metabolism.** The purine metabolism shows the first step of *de novo* purine synthesis mediated by the enzyme 5′-phosphoribosyl-1-pyrophosphate (PRPP) synthase, and the second step by PRPP amidotransferase. PRPP from adenosine-triphosphate (ATP) and ribose-5-phosphate is catalyzed by PRPP synthase. PRPP amidotransferase converts PRPP to ribosylamine-5-phosphate (PRA). This is then catalyzed by a series of enzymes to produce hypoxanthine nucleotides (IMP), which in turn produce adenosine monophosphate (AMP) and guanosine monophosphate (GMP). Xanthine oxidase (XO) converts hypoxanthine to xanthine and xanthine to uric acid. Meanwhile, the salvage pathway mediated by hypoxanthine phosphorybosyltransferase (HPRT) and adenine phosphorybosyltransferase (APRT). The enzyme HPRT salvages hypoxanthine to IMP and guanine to GMP. In a similar salvage pathway, APRT converts adenine to AMP.

has dual inhibition against xanthine oxidase (XOD) and URAT1 and can reduce UA levels (*Dong et al., 2019*). A selective URAT1 inhibitor, RDEA3170, is currently in Phase II clinical trials to treat gout and hyperuricemia (*Sun et al., 2021*).

Hyperuricemia can be caused by excessive urate synthesis or insufficient renal excretion. However, not all patients with hyperuricemia necessarily cause gout. MSU crystallization is a prerequisite for gout attacks. Despite the fact that hyperuricemia is the main factor that leads to the crystallization of MSU, other factors such as temperature, pH, connective tissue conditions, and proteins can also alter the crystallization of MSU (*Zhang, 2021*). Conversely, not all people with gout always suffer from hyperuricemia (*Lee et al., 2020*). In certain states (fasting) and with certain medicines (thiazide diuretics, favipiravir), SUA concentration can increase (*Zhang, 2021*). Furthermore, renal insufficiency patients have difficulty maintaining normal urate levels due to damaged renal urate transport proteins (*Sivera, Andres & Dalbeth, 2022*).

### Inflammation caused by urate deposition

There is a possibility that MSU crystals will form when the saturation concentration of UA is reached in the blood. During the early stages of gout, MSU causes neutrophils to produce an excessive amount of inflammatory cytokines, which causes a flare-up of symptoms. Furthermore, MSU can also induce phagocytes to release many mediators, including the cytokines interleukin-6 (IL-6), tumor necrosis factor-$\alpha$ (TNF-$\alpha$), interleukin-1$\beta$ (IL-1$\beta$), growth-related oncogene $\alpha$ (GRO $\alpha$)/CXCL1, the $C-X-C$ chemokines (IL-8)/CXCL8, and myeloid-related proteins S100A8, S100A9 (*Rousseau et al., 2017*; *Hemshekhar et al., 2020*), continuously drive the inflammatory response. IL-1$\beta$ is well established as the primary trigger for inflammation (*Pillinger & Mandell, 2020*). The process of IL-1$\beta$ release is crucial: One is that inflammatory signals induce gene expression and synthesis of the pro-IL-1$\beta$; The second is exogenous ATP or reactive oxygen species (ROS), which causes the NACHT, LRR, and PYD domains-containing protein 3 (NLRP3) inflammasome activation and secretion of IL-1$\beta$ (*Wu et al., 2020*).

MSU crystals recognize entry into macrophages and activate the NLRP3 inflammasome. NLRP3 inflammasome activates cysteinyl aspartate specific proteinase-1 (caspase-1) that cleaves pro-IL-1$\beta$ to produce IL-1$\beta$ (*Wu et al., 2019a*). Further, extracellular ATP activates P2X7 receptor 7 (P2X7R) in macrophages and sets off potassium ion outflow, causing the NLRP3 inflammasome to release IL-1$\beta$ (*Alessandra Piccini et al., 2008*; *Marinho et al., 2020*). IL-1$\beta$ promotes the release of various inflammatory mediators, including TNF-$\alpha$, interleukin-1(IL-1), IL-6, and IL-8. As for the other mediators, like prostaglandin-E2, leukotrienes, and ROS, they can also be released in large amounts (*Szekanecz et al., 2019*; *Luo et al., 2020*). Polymorphonuclear leukocytes are recruited when chemokines and adhesion factors are released, resulting in systemic inflammation (*Rousseau et al., 2017*). S100A9 could enhance neutrophils induced by MSU and increase IL-8/CXCL8, IL-1, and ROS production (*Rousseau et al., 2017*). A series of inflammatory reactions revealed that IL-1$\beta$ is the core of gout. In the clinical setting, IL-1$\beta$ blockers (*Otani et al., 2020*; *Szekanecz et al., 2019*) are widely used to relieve inflammation among patients with gout. The availability of IL-1$\beta$ inhibitors, however, is limited to certain populations. The current first-line treatment for gout attacks is anti-inflammatory treatment with corticosteroids, NSAIDs, or colchicine (*Richette et al., 2017*). In gout patients, experts recommend controlling pain and inflammation swiftly as a short-term goal. However, serum urate-targeted therapies seem to be the most appropriate option for the long-term management of gout patients. Considering that gout is a disease caused by the deposition of urate crystals, urate-lowering therapy (ULT) is the key to preventing gout recurrence. Reduced urate levels result in the crystals gradually dissolving, thus improving gout sufferers' outcomes. In general, ULT options consist of reducing urate synthesis (allopurinol and febuxostat), increasing renal urate excretion (benzbromarone and probenecid), and using uricase (*Sivera, Andres & Dalbeth, 2022*). Asymptomatic hyperuricemia therapy has attracted attention in addition to the therapeutic management of gout patients. This attention may be due to the potential impact of high SUA levels on cardiovascular disease and the kidney (*Brucato, Cianci & Carnovale, 2020*). As such, gout management requires both ULT and inflammatory inhibitors in combination.

## Proteomic study of hyperuricemia and gout
### Plasma proteomics

There are some proteomic studies of hyperuricemia and gout (Table 1). While detecting very low abundance proteins in blood samples is difficult, they are the most frequently collected samples. Proteomics relies on MS to identify proteins in 2-dimensional gel electrophoresis (2-DE) samples. In many cases, time-of-flight (TOF) analyzers are used in conjunction with matrix-assisted laser desorption/ionization (MALDI). Several studies have used 2-DE technology in combination with MALDI-TOF/MS to analyze serum samples to understand the protein changes caused by patients with hyperuricemia. Fei's study showed that Uyghur patients with hyperuricemia had higher serum complement C3, haptoglobin, complement C4, and apoprotein A1 (apo A1) levels compared to normal people (*Fei et al., 2012*). Following up on this study, *Fei et al. (2013)* used the same technique to analyze serum from Han Chinese patients with hyperuricemia. There are several proteins with differential expression that are largely the same and can influence hyperuricemia development. Two studies in Xinjiang, China, found different protein profiles in Han and Uyghur hyperuricemia patients. It seems that this finding could indicate that there may be different protein expression patterns within diverse ethnic groups (*Fei et al., 2012*; *Fei et al., 2013*). Additionally, a study of patients with acute gout (AG) showed that plasma concentrations of apolipoprotein A-I (apo A-I) had a proportional relationship with UA. That is, there was a high expression of apo A-I in the plasma of the patients (*Chiang et al., 2014*). These detected proteins showed a link between hyperuricemia and high-density lipoprotein (HDL) components.

An HDL molecule prevents thrombosis, has anti-inflammatory and complement-activating properties, as well as acts as an antioxidant. As a major part of HDL, Apo A-I plays a role in cholesterol transport and anti-atherosclerotic effects. Increased levels of apo A-I during acute gouty arthritis (AGA) may lead to spontaneous remission (*Georgila, Vyrla & Drakos, 2019*). HDL has been shown as a modulator of lipid metabolism and influences the development of cardiovascular disease and diabetes (*Gordon & Remaley, 2017*; *vander Vorst, 2020*). Several animal studies seem to clarify the anti-inflammatory mechanism of apo A-I. In an experiment, a rat model of pregnancy-induced insulin resistance enhanced insulin sensitivity in adipose tissue after infusion of apo A-I. Both TNF-$\alpha$ and IL-6 in rat plasma were decreased, and apo A-I could inhibit inflammation (*Wu et al., 2019b*). Like apo A-I, apo A1 treatment induces a significant reduction in macrophage chemotaxis and inhibition of monocyte recruitment *in vivo*, exerting an anti-inflammatory effect (*Iqbal et al., 2016*). Thus, apo A-I inhibition may contribute to the development of atherosclerosis (AS) through a mechanism of chronic inflammation. Investigators found that an apo A-I/HDL inducer (rvx208) reduced total atherosclerotic volume in patients with coronary artery disease in a phase II trial (*Ghosh et al., 2017*). In diseases such as AS, HDL and its related components may influence disease progression. Apolipoprotein expression in HUA patients is closely associated with cardiovascular disease development. Because of the complex and abundant variety of differential serum proteins, we can only speculate about their relevance to hyperuricemia and gout. Further research is needed to clarify the direct relationship between these study indicators and hyperuricemia and gout.

Wu and You (2023), *PeerJ*, DOI 10.7717/peerj.14554

**Table 1  Overview of proteomics studies in hyperuricemia and gout.** We performed a detailed analysis of proteomic studies of hyperuricemia and gout including sample types, methods used, sample sizes, protein involved pathways, and key proteins.

| Reference | Study (Year) | Sample | Technique | Study size | The main participation way | Proteins | Key proteins |
|---|---|---|---|---|---|---|---|
| Fei et al. (2012) | Fei et al. (2012) | Serum | 2-DE and MALDI-TOF-MS | Hyperuricemia:15; Control group:15 | – | C3, C4, haptoglobin, and apo A1. | C3, C4, haptoglobin, apo A1. |
| Fei et al. (2013) | Fei et al. (2013) | Serum | 2-DE and MALDI-TOF-MS | Hyperuricemia:15; Control group:15 | – | C3, haptoglobin, $\alpha$1 -antitrypsin, and apo L1. | C3, haptoglobin, $\alpha$1-antitrypsin, and apo L1. |
| | | Plasma | 2-DE and MS | Male acute gouty patients:80; Male controls:129 | – | Fibrinogen gamma, transthyretin, pre-serum amyloid P component, CRP, and apo A-I. | Apo A-I |
| Chiang et al. (2014) | Chiang et al. (2014) | Synovial fluid | LC/MS/MS | Male acute gouty patients:24 | – | Fibrinogen, ORM1and2, angiotensinogen, alpha 2-macroglobulin, apo A-I, apo D, and apo H. | Apo A-I |
| Shen et al. (2021a); Shen et al. (2021b) | Shen et al. (2021a); Shen et al. (2021b) | Plasma | HPLC-MS/MS | Gout patients:8; Gout patients combined with renal injury:8 | These proteins participated in innate immune response, platelet degranulation, protein hydrolysis, and the classical activation pathway of complement. | In the normal control group and gout group: 32 proteins; In the gout and gout with renal injury group: 10 proteins; Correlated with SUA: GSN, S100A8, S100A9, ORM2, and ANXA1. | GSN, S100A8, S100A9, ORM2, and ANXA1. |
| Chen et al. (2021a); Chen et al. (2021b) | Chen et al. (2021a); Chen et al. (2021b) | Plasma | iTRAQ-PRM | AG: 8; RG: 7; AHU: 7; Healthy controls: 8 | The most representative pathway was peroxisome proliferator activated receptor signaling pathways and alcoholism pathway. The complement and coagulation cascades is one of the main functional pathways. | Eleven differentially expressed proteins such as Histone H2A, Histone H2B, THBS1, Inter-alpha-trypsin inhibitor heavy chain H4, ORM1, Multimerin-1, Myeloperoxidase, CA1, albumin, C8B and C2. | Histone H2A, Histone H2B and THBS1. |
| Huo et al. (2021) | Huo et al. (2021) | Urine | Label-free LC −MS/MS | HUA:26; Healthy controls: 25 | Including the processes for insulin receptor recycling and lipid metabolism. | In HUA samples: 11 proteins were found decreased and 2 proteins were found increased. | VATB1, CFAD and APOC3. |
| Chiu et al. (2015) | Chiu et al. (2015) | Pouch membranes | iTRAQ and LC-MS/MS | Negative control pouch membranes:2; Inflamed pouch membranes:3 | The identified proteins are primarily related to immune-related complement system and the tricarboxylic acid cycle. | Alternative complement pathway: C3, C8 $\beta$, C9, complement component factor I, CFAD, complement factor B, clusterin, integrin $\alpha$M, and integrin $\beta$-2. Related to NALP3 inflammasome: CRAMP and S100A9. | Two upregulated proteins, S100A9 and CRAMP. |
| Huang et al. (2022) | Huang et al. (2022) | Synovial Fluid-Derived Exosomes | TMT labeled LC-MS/MS | Gout:42; Rheumatoid arthritis:30; Axial spondyloarthritis:10; Osteoarthritis:18 | These proteins were significantly involved in complement and coagulation cascades, acute-phase response and citrate cycle in gout. | In gout: Sixty-nine differentially expressed proteins were found. Twenty-five proteins were found highly expressed in gout uniquely, lysozyme C, ORM1, lactotransferrin, S100A9, and myeloperoxidase included. | – |

Wu and You (2023), *PeerJ*, DOI 10.7717/peerj.14554

**Table 1** (*continued*)

| Reference | Study (Year) | Sample | Technique | Study size | The main participation way | Proteins | Key proteins |
|---|---|---|---|---|---|---|---|
| Kaneko et al. (2012) | Kaneko et al. (2012) | Urinary stone | SDS-PAGE and LC-MS/MS | Gout or hyperuricemia patients:15; Urinary stones:17 | Some of these proteins should play an essential role in the early stage of stone formation. | In CaOx stones, osteopontin, uromodulin, albumin, protein Z, prothrombin, protein S, hemoglobin and histone H4 were identified. In UA stones, uromodulin, albumin, hemoglobin, calgranulins and immunoglobin G fragments were detected. In CaOx stones and UA stones: albumin, hemoglobin, uromodulin, calgranulin A and B, and histone H4 were detected. | In CaOx stones: osteopontin, prothrombin, protein Z and protein S were often identified. In UA stones: IgG fragment was detected characteristically. |
| Kaneko et al. (2014) | Kaneko et al. (2014) | Gouty tophus | SDS-PAGE and LC-MS/MS | Patients:1; Tophus:1 | Many proteins relevant to inflammation and host defense were identified. | Proteomic analysis identified 134 proteins from the tophus as matrix proteins. | Immunoglobulins |
| Kaneko et al. (2018) | Kaneko et al. (2018) | A urinary stone with two layers | SDS-PAGE and LC-MS/MS | Patients:1; Urinary stone:1 | 14 of 51 were functionally categorized as cell adhesion and cytoskeleton proteins and 7 as metabolism-related proteins. There were 5 defense proteins and 5 plasma proteins. | In COM part: 48 proteins (non-muscle myosin heavy chain, uromodulin, coagulation factor II, protein S, protein Z, apo E, and others); In UA part: 7 proteins (S100A8, hemoglobin, albumin, and others); In the interface: 4 proteins (dermcidin, coagulation factor II, keratin 1, and osteopontin isoform OPN-b). | Proteins relevant to cell adhesion, self-defense, and plasma commonly play a major role in the generation of both COM and UA part. |

**Notes.**

Abbreviations: 2-DE, two-dimensional gel electrophoresis; MALDI-TOF-MS, matrix-assisted laser desorption ionization-time of flight mass spectrometry; C, complement; apo, apolipoprotein; LC, liquid chromatography; MS, mass spectrometry; CRP, C-reactive protein; ORM1and2, Alpha-1-acid glycoprotein 1and2; HPLC, high-performance liquid chromatography; SUA, serum uric acid; GSN, gelsolin; S100A8, Protein S100-A8; S100A9, Protein S100-A9; ANXA1, Annexin A1; iTRAQ-PRM, isobaric tag for relative and absolute quantification- parallel reaction monitoring; THBS1, Thrombospondin-1; HUA, asymptomatic hyperuricemia; VATB1, V-type proton ATPase subunit B kidney isoform; CFAD, Complex factor D; APOC3, Apolipoprotein C3; NALP3, NACHT, LRR, and PYD domains-containing protein 3; CRAMP, Catherine-related antimicrobial peptide; TMT, tandem mass tag; SDS-PAGE, sodium dodecyl sulfate-polyacrylamide gel electrophoresis; CaOx or COM, calcium oxalate.

Recent proteomic studies show that gout and hyperuricemia trigger the complement system in the body. *Shen et al. (2021a)* detected differential plasma proteins using label-free quantitative proteomics in simple gout patients, gout with renal damage patients, and non-gout patients. As mentioned earlier, these three groups of detected proteins are thought to be involved in natural immune responses, complement activation, *etc.* More importantly, among these proteins, inflammation factors gelsolin, S100A8, S100A9, Alpha-1-acid glycoprotein 2, and Annexin A1 were most significantly associated with SUA. It may therefore be possible to develop a combinatorial model of diagnosing gout disease as a result of this research. Many researchers have recently switched from 2-DE to MS-based quantification tools. Among the most widely used quantitative proteomics techniques are iTRAQ and TMT. Chen and co-workers developed an iTRAQ and parallel reaction monitoring method to analyze plasma protein profiles of AG, remission of gout (RG), and AHU patients (*Chen et al., 2021a*). Researchers found 11 essential proteins, most of which regulate cytokines or enhance inflammation in gout. It may be possible to distinguish AG, RG, AHU, and healthy people based on Histone H2A, Histone H2B, and Thrombospondin-1 (THBS1). THBS1, a glycoprotein found in the extracellular matrix, is capable of bridging cell interactions and promoting inflammation (*Rico et al., 2008*). As chronic inflammation regresses, THBS1 expression maintains tissue homeostasis and reduces inflammation (*McMorrow et al., 2013*). Patients who are suffering from AG may benefit from the regression of inflammation as a result of THBS1. A biomarker may provide insight into the diagnosis of gout based on the discovery of these proteins. From the proteomic characteristics of this study, the team found that complement and coagulation cascades were one of the main functional pathways in the gout process. The interaction between the complement and coagulation systems is likely to influence the development of inflammation in some ways.

The activation of NLRP3 can lead to various pathological inflammatory diseases, including As, gout, and type 2 diabetes (T2DM). Complement and coagulation systems may be closely involved in activating NLRP3. An arousal of the complement system stimulates platelet activation and promotes a coagulation process. However, thrombosis can activate the complement system and trigger an inflammatory response further (*Oncul & Afshar-Kharghan, 2020*). During protein composition analysis, we found that serum samples from gout patients expressed higher levels of complement fractions. In addition to being an essential component of innate immunity, the complement system is also involved in preventing infections, among other things. Component C5a of the complement system is a potent pro-inflammatory mediator, increasing the activation of monocytes and neutrophils. MSU can induce IL-1 $\beta$ and release inflammatory cytokines by activating complement system component C5a (*Khameneh et al., 2017*; *Yu et al., 2019*). For this reason, some investigators have considered complement antagonists as a means of controlling inflammation. While complement antagonists can control gout inflammation relatively well, multiple factors affect the inflammation process. Therefore, C5a antagonists may combine with IL-1 $\beta$ blockers as a therapeutic target for treating inflammatory diseases.

According to the above studies, most studies on hyperuricemia or gout studied proteomics separately. As a result, finding biomarkers to differentiate hyperuricemia from gout is challenging. To date, this is the only proteomic study that distinguishes between hyperuricemia and gout. According to Chen's study, four proteins were significantly more expressed in AG patients than in the AHU group. Gout patients may experience inflammatory episodes as a result of the upregulation of serum acute phase reaction protein. Compared to the AHU group, some proteins were expressed at lower levels in the AG group, multimerin-1 being the lowliest expressed (*Chen et al., 2021a*). According to the results of quantitative proteomics studies conducted in patients with gout and hyperuricemia, the expression of certain proteins is different when a single protein is targeted. Secondly, there is a difference in the general trend of protein types. The major difference between hyperuricemia and gout is the development of MSU crystals that cause gout attacks. The presence of significantly expressed inflammatory proteins in gout patients suggests that the inflammatory mechanism deserves our attention (*Shen et al., 2021a*; *Qiu et al., 2021*). As a final observation, patients with gout showed an even more profound association with complement and coagulation cascade pathways (*Shen et al., 2021a*; *Chen et al., 2021a*; *Huang et al., 2022*). In summary, multiple biomarkers can provide insights into disease mechanisms based on their expression and functions through a holistic approach to identifying diseases.

### Urine proteomics

Among the noninvasive methods of collecting samples, urine is one of them. It has certain advantages to studying urine samples at the protein level. A study led by Huo compared urine protein metabolism in healthy individuals and patients with HUA (*Huo et al., 2021*). As a result of the enrichment analysis of these differentially expressed proteins, plausible pathways for insulin receptor recycling and lipid metabolism have been identified. Among them, V-type proton ATPase subunit B kidney isoform and Complex factor D (CFAD or adipsin) affect insulin levels in patients with hyperuricemia. Hypertriglyceridemia in patients with hyperuricemia may be associated with apolipoprotein C3 (APOC3). Furthermore, the plasma of gout patients also has abnormal levels of apolipoproteins (APOC4, APOD) (*Shen et al., 2021a*). As part of our investigations, we analyzed some proteomic studies about patients with metabolic abnormalities, diabetes, and obesity. According to some studies, pro-inflammatory biological and lipid markers are directly responsible for obesity in obese patients (*Doumatey et al., 2016*). Adipose tissue can secrete some adipokines that cause the chronic inflammatory response to obesity. Indeed, a fraction of apolipoproteins can inhibit the inflammatory pathways of adipose tissue (*Wang et al., 2018*). As a result, APOC3 or CFAD may be an additional risk factor for the development of metabolic diseases.

APOC3 inhibits lipoprotein lipase and hepatic lipase activity, thereby reducing triglyceride-rich lipoproteins and residual uptake in the liver (*Norata et al., 2015*). Triglyceride metabolism and the prevalence of T2DM are closely related to APOC3. Several proteomic studies have linked apolipoprotein expression to triglyceride metabolism in diabetes and obesity (*Doumatey et al., 2016*). However, it is unclear whether APOC3

alone regulates triglyceride concentrations in patients with hyperuricemia. APOD is a class of proteins that transports lipids, and arachidonic acid (AA) is its common ligand. APOD exerts neuroprotective and anti-inflammatory effects by stabilizing AA at the cell membrane (*Rassart et al., 2020*). In a cohort study, the authors examined APOD expression in three intra-abdominal adipose tissues, including omental, mesenteric, and round ligament (RL). In particular, with high levels of APOD in the RL depot, women exhibited lower plasma insulin levels and lowered circulating pro-inflammatory PAI-1 and TNF-$\alpha$ levels (*Desmarais et al., 2018*). APOD has been shown to correlate with obesity and insulin resistance. These findings show that patients with hyperuricemia and gout are associated with lipid metabolism disorders that are capable of promoting diabetes and hyperlipidemia. Adipocytes secreted fat factors such as CFAD that participated in activating complement component C3a and could protect $\beta$ Cells. At the same time, complement C3a can enhance $\beta$ Cellular insulin secretion (*Tafere et al., 2020*). Hence, it is unsurprising that adipsin can be used as a T2DM marker. There was a lower risk of diabetes when adipsin levels were lower than BMI (*Gomez-Banoy et al., 2019*). In addition, it has been demonstrated that adipocytes can affect inflammatory arthritis. Adipsin is an intermediate mediator, regulating the neutrophil-induced inflammatory response (*Li et al., 2019*). Thus, obesity can also result in an inflammatory response in patients with gout. The discovery of CFAD in patients with hyperuricemia may affect the prevalence of T2DM. APOC3 and CFAD may serve as potential targets as related factors.

Observations of hyperuricemia and metabolic diseases have shown that apolipoproteins and adipokines have a correlation with disease progression. Despite the fact that there was only one urine proteomics study on hyperuricemia, the results suggest it is a risk factor driving metabolic diseases. The lack of urine proteomics studies may result from the characteristics of urine proteins and the timing of urine collection (*Sanchez-Juanes & Gonzalez-Buitrago, 2019*).

### Synovial fluid proteomics

Several studies have investigated the proteome profiles of SF-derived exosome samples aiming to determine the impact on gout progression. Researchers have made the first ingenious use of animal gout models to explore the protein profiles expressed during gout attacks. *Chiu et al. (2015)* injected sodium urate crystals into the murine air pouch (which resembles the synovial membrane) to induce inflammation. Based on proteomic analysis, the differentially expressed proteins participated in the alternative complement pathway. The results suggested that the levels of Catherine-related antimicrobial peptide (CRAMP) and S100A9 were positively correlated with MSU. According to the authors, NALP3 activation in late gout is associated with the upregulation of S100A9 and CRAMP proteins. It is possible to use these differential proteins as potential therapeutic targets in the future for AG. Since MSU crystal deposits in SF or tophi are the gold standard for microscopy in gout patients, several authors have started to analyze SF samples from gout patients. By combining TMT and LC-MS/MS, researchers have quantified more proteins, even those with low expression levels. In a study analyzing the proteomics of SF-derived exosomes, 69 proteins were found to be differentially expressed in gout patients. There

are 25 highly unique expressed proteins, including lysozyme C, alpha-1-acid glycoprotein 1, lactotransferrin, S100A9, and myeloperoxidase. They may play a role in neutrophil degranulation, prion diseases, as well as complement and coagulation cascade complement cascades (*Huang et al., 2022*). Similarly, *Qiu et al. (2021)* found that differentially expressed genes in the male gout group were predominantly enriched in the inflammatory response, S100 protein binding. Therefore, the inflammatory response associated with differentially expressed proteins plays a vital role in gout patients.

Several studies have implicated S100A8 and S100A9 in gout inflammation. There is an increase of at least 20 times the concentration of S100A8/A9 in SF local sites compared to serum. The selection of S100A8/A9 better reflects the relevance of disease activity than conventional biomarkers (*Pruenster et al., 2016*). In addition, S100A8/S100A9 concentrations may predict the response to drug therapy in some patients with inflammatory arthritis (*Choi et al., 2015*). S100A8 and S100A9 are considered to be the most promising biomarkers (*Austermann, Zenker & Roth, 2017*). In the same way as S100A8 and S100A9, CRAMP is a component of innate immunity that prevents pathogen infection, damage, and repair and is involved in chemotaxis and inflammatory response activation *GHGudmundsson (1999)*. Interestingly, CRAMP and LL-37 have some similarities in structure and function. CRAMP and LL-37 belong to antimicrobial peptides (AMPs, also known as host defense peptides). Human cathelicidin protein is LL-37, while CRAMP is its functional homologue in mammalian cells (*Kang et al., 2020*). Under inflammatory conditions, the antimicrobial peptide LL37 can be produced by human local SF to resist intra-articular inflammation (*Varoga et al., 2005*; *Hitchon et al., 2019*). Although the specific physiological role of LL37 in hyperuricemia and gout is unknown, it has been confirmed that LL37 can affect the progression of inflammation. These proteins may provide new targets for inflammatory drug therapy in gout patients.

An examination of the crystals of MSU present in SF under the microscope is the gold standard for confirming gout in the patient. As a result, SF proteomics is more likely to reflect differential proteins specific to gout patients. As we suspected, the results of the study above also corroborated our suspicions.

### *Urinary stone proteomics*

The presence of hyperuricemia is associated with renal damage. A decrease in renal UA excretion is a common cause of hyperuricemia and the main mechanism of hyperuricemia in most cases of gout. In light of the fact that hyperuricemia can result in kidney damage, such as the formation of urinary stones, we investigated urinary stone proteomics in patients with hyperuricemia or gout to identify unique biomarkers. In proteomics, the most commonly used technique is LC-MS/MS. Proteins separated by sodium dodecyl sulfate-polyacrylamide gel electrophoresis can be analyzed with LC-MS/MS to obtain high-throughput amino acid sequence information. *Kaneko et al. (2012)* investigated the association between matrix proteins and calcium oxalate (CaOx) stone or UA stone formation by studying matrix proteins in 17 urinary stones (patients with gout and hyperuricemia). Proteins such as urinary regulatory proteins and albumin are frequently detected in stones. CaOx stones, however, contain bone-bridging proteins,

thrombospondin, protein S, and protein Z, while UA stones contain IgG fragments. Therefore, these proteins may contribute to the formation of urinary stones, inhibiting or contributing to their aggregation. Some proteins associated with cell adhesion, cytoskeleton, self-defense, and plasma usually play a significant role in stone formation (*Kaneko et al., 2018*). *Jou et al. (2012)* also discussed the relationship between proteins involved in the phospholipid or fatty acid pathway and stone formation.

Furthermore, interactions in the study of urinary stones are not necessarily limited to protein, but crystal-protein interactions are also important for forming kidney stones. An analysis of crystal-protein interaction studies showed that altering protein coating on MSU and M-CPPD can inhibit or promote crystal-induced IL-1 $\beta$ inflammatory processes. Surface IgG stimulates inflammation, while HLD, apolipoprotein A, and apolipoprotein E inhibit it (*Ortiz-Bravo Jr & Schumacher, 1993*; *Renaudin et al., 2019*). The anti-inflammatory activity of apolipoprotein B in mouse airbags has been implicated in previous studies (*Ortiz-Bravo Jr & Schumacher, 1993*). The changes in the inflammatory properties of the crystal by protein coating may provide a new method for treating inflammation in gout patients. Some researchers found that proteins detected in urinary stones are usually involved in immune and inflammatory responses, suggesting a role for inflammation in stone formation. Therefore, we recommend that this is one of the primary mechanisms by which UA causes kidney damage. For example, in work published in 2012, most of the proteins tested were involved in the inflammatory and complement systems, followed by the interleukin-6 signaling pathway (*Jou et al., 2012*). It is always known that factors such as myeloid-associated proteins and complement membrane attack complexes are mediators of acute crystal-induced inflammation. Myeloid-associated proteins (S100A8 and S100A9) were detected at the highest frequency among these inflammatory proteins (*Kaneko et al., 2014*). In addition, some studies have found elevated expression of S100A8 and S100A9 in renal tissues of patients with kidney stones but undetectable in the urine of healthy people (*Yang et al., 2021a*). Therefore, with diagnostic significance, S100A8 and S100A9 can serve as effective biomarkers for urinary stones.

There is a possibility that urinary stones may form in patients with renal injury due to insufficient urate excretion leading to the deposition of MSU crystals. Structural studies that target urinary calculi can provide clues about the prevention of renal injury through proteomic studies. In spite of this, it remains questionable whether these differential proteins can specifically be expressed in patients with hyperuricemia or gout, and more research is needed to determine this. It is noteworthy that the special protein coating of the outer layer of the stone can inhibit the inflammatory response, giving us a new therapeutic option.

## Metabolomics of hyperuricemia and gout
### Amino acid metabolism
Metabolomics analyzes the molecular composition of a sample by using NMR or MS, while both liquid chromatography and gas chromatography are used to separate metabolites. As shown in Fig. 2, hyperuricemia and gout are associated with various metabolic disorders. Several studies have shown that amino acid metabolism is highly associated
with hyperuricemia and gout. An investigation of the plasma-free amino acid profile (PFAA) of gout patients by *Mahbub et al. (2017a)* revealed that some amino acids were expressed differently. Researchers found positive correlations between alanine, isoleucine, leucine, phenylalanine, tryptophan, valine levels, and gout. In contrast, the levels of glycine and serine were negatively associated with gout. Similarly, patients with gout also show abnormal expression of metabolites of purine metabolism and branched-chain amino acid metabolism. In the serum, valine, leucine, and phenylalanine levels were elevated (*Huang et al., 2020*). A disorder of amino acid metabolism may be present in patients with gout. According to a cross-sectional study, differences in PFAA concentrations existed with or without HUA, and PFAA and UA levels correlated with the risk of lifestyle-related diseases (*Mahbub et al., 2017b*). Following a different methodological approach, *Wang et al. (2020a)* studied the relationship between hyperuricemia and amino acids. UA levels in hyperuricemia patients were affected by cysteine, glutamine, phenylalanine, and threonine. Since glycine (*Joseph, Baggott & Tamura, 2007*) and glutamine (*Otani et al., 2020*) are essential amino acids that are required for the *de novo* synthesis of purine, the change in amino acids will affect the production of UA. In addition to affecting changes in UA levels, amino acid metabolism disorders can also contribute to the progression of other metabolic diseases. In addition to this, these amino acids are a reflection of the dietary habits of hyperuricemia patients. It should be noted that glutamine is more common in meat products, while phenylalanine and threonine are more common in bean products. A thorough understanding of the purine production and the dietary consumption of hyperuricemia and gout can assist in preventing these diseases.

The transition stage of hyperuricemia and gout onset is ambiguous. The early symptoms of AHU are often ignored, resulting in the loss of treatment time in the disease process. A metabolomic approach reveals disturbing pathways in serum HUA and gout patients. Compared to the three metabolic pathways found in hyperuricemia patients, this study showed a more severe metabolic disorder in gout patients and an increase in glutamine that was only seen in gout patients (*Zhang et al., 2018*). In addition, significant differences in arginine metabolism occurred in both hyperuricemia and gout patients. Shen and his colleagues suggest that arginine anabolic metabolism can contribute to the progression of inflammation in gout patients. It may be possible to distinguish between the two diseases using arginine as a biomarker (*Shen et al., 2021b*). *Luo et al. (2018)* also conducted an LC-MS/MS analysis of AG and AHU's plasma amino acids profile. As gout patients have severe metabolic disorders, it is appropriate for them to consume more glycine as a result of their metabolic disorders. When serum UA levels are similar, L-isoleucine, L-lysine, and L-alanine may be potential markers to distinguish AG from AHU. Interestingly, when the amino acid content changes, the precipitation of MSU at the limb end of AG patients is more accessible. Most likely because changes in the plasma amino acids affect the production of MSU, this may be the reason why this happens. *Oshima, Shiiya & Nakamura (2019)* attempted to treat patients with mild hyperuricemia using supplementation with a mixture of glycine and tryptophan. The results of this study indicated that this supplementation prevented MSU crystal deposition by increasing UA solubility and elevating urinary pH. Different amino acid metabolic abnormalities are present in patients with hyperuricemia

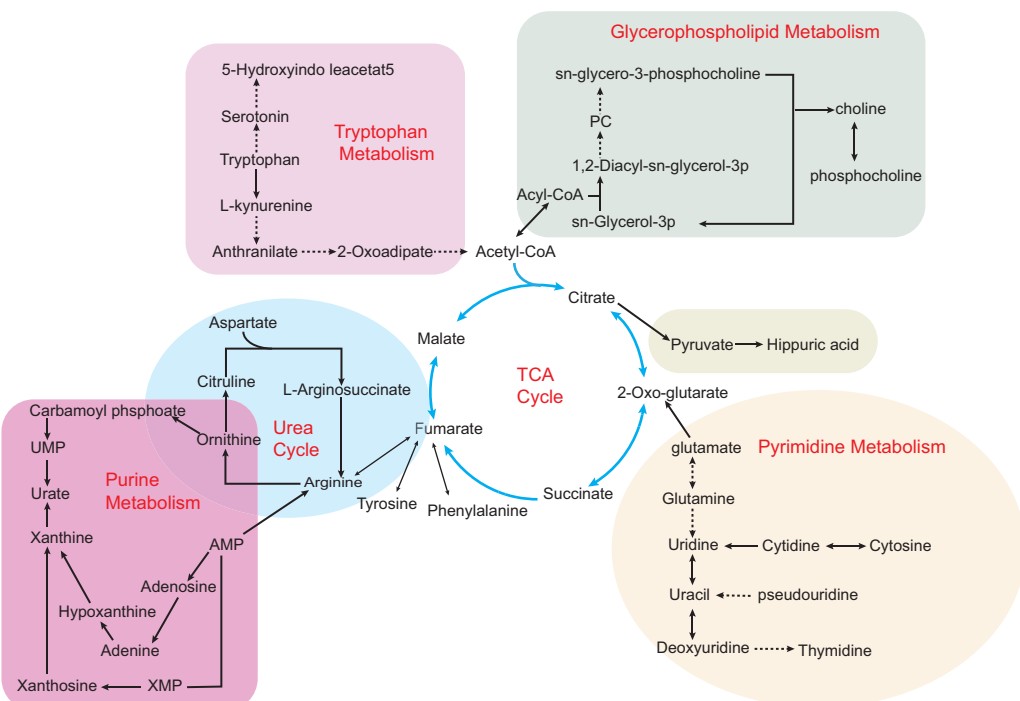

**Figure 2 Some metabolic disorders pathways in hyperuricemia and gout.** The network is based on the metabolic pathways and biomarkers mainly affected by hyperuricemia and gout. Metabolic processes that cause abnormalities include the tricarboxylic acid (TCA) cycle, purine metabolism, pyrimidine metabolism, glycerophospholipid metabolism, and gut microbial metabolism. Also, various amino acid metabolism disorders are common, such as tryptophan metabolism, ornithine and arginine biosynthesis in the urea cycle. Key intermediates of the TCA cycle such as citric acid, 2-ketoglutarate and succinic acid metabolites build up the entire metabolic network, making the TCA cycle an intertwining point of metabolic pathways.

and gout. Besides reflecting the disease process, amino acid metabolism also plays a role in the differential diagnosis. Therefore, establishing relevant amino acid biomarker models is essential to the future of hyperuricemia and gout.

Hyperuricemia and gout may express different amino acids. Therefore, amino acids may be promising biomarkers to differentiate between patients with hyperuricemia and gout. Amino acid types not only reflect the metabolic processes *in vivo* in hyperuricemia and gout but also are closely related to dietary habits and other aspects. However, identifying amino acid types in gout and hyperuricemia cannot be accomplished by simply identifying a single metabolite. We need to recognize that amino acid metabolism is influenced by confounding factors such as exogenous and endogenous sources. Biomarker models of the single or multiple amino acids that make up the disease are not yet possible.

### Other metabolic pathways

The purine metabolic pathways are also closely related to the development of hyperuricemia. UA is the end product of purine metabolism, and when purine nucleotide synthesis is disturbed, serum UA concentration increases (*Li et al., 2018*). It is then followed by the

formation of urate crystals, which causes gout inflammation. Similar to purine metabolism, other metabolic pathways influence hyperuricemia and gout in one way or another. In both untargeted and lipid-targeted metabolomics studies, serum samples from patients with hyperuricemia have detected metabolites related to glycerophospholipid metabolism and sphingolipid metabolism (*Qin et al., 2021*; *Qin et al., 2022*). *Li et al. (2018)* also found significant differences in glycerophospholipid metabolism between gout patients and controls. Therefore, researchers are gradually focusing on the relevance of lipid metabolism to patients with hyperuricemia and gout. The conversion of some metabolites, such as saturated fatty acids, involves producing inflammatory factors that may cause an inflammatory response in HUA patients (*Berg, Seyedsadjadi & Grant, 2020*). Obesity and hyperlipidemia have been shown to be risk factors for hyperuricemia (*Lai et al., 2021*). According to research, BMI, waist circumference, and body fat percentage are associated with hyperuricemia (*Rivera-Paredez et al., 2019*). We should pay attention to the fact that glycerophospholipid metabolism is strongly affected by the food we consume. According to a prospective study, consumption of ultra-processed food (UPF) (*Zhang et al., 2021*) and a Western diet are related to the risk for hyperuricemia. In terms of composition, UPF and Western diets generally include a large amount of fat, saturated fat, sugar, and salt. These findings suggest that hyperuricemia has a disorder of lipid metabolism, and a diet with high-fat content will increase the risk of hyperuricemia.

In addition, the rich metabolic pathways in patients with hyperuricemia and gout involve carbohydrate and energy metabolism (*Huang et al., 2020*; *Zhang et al., 2018*). It is well known that the body derives its energy mainly from the tricarboxylic acid (TCA) cycle. Succinic acid and fumaric acid are important intermediate metabolites of the TCA cycle, and the decline of both substances in gout patients reflects excessive energy depletion (*Li et al., 2018*). In particular, glucose levels are higher in gout patients, suggesting that metabolic disorders are more severe in gout patients than in hyperuricemia patients. Interestingly, a noteworthy study identified primary bile acid biosynthesis as a possible new metabolic pathway. *Zhong et al. (2021)* studied the serum metabolic profiles of metabolite extracts from gout patients, and urate and bilirubin are reliable metabolites in gout patients. Significantly different metabolites are mainly involved in primary bile acid biosynthesis, purine metabolism, and glycerophospholipid metabolism. Bile acids can affect UA levels in mice by interfering with orphan nuclear receptor peroxisome proliferator-activated receptor alpha activation of the XOD gene (*Kanemitsu et al., 2017*). Thus, bilirubin may be a potential biomarker influencing changes in UA levels. While new metabolic pathways are constantly being discovered to influence the development of hyperuricemia and gout, the metabolic pathways intersect with each other, thus forming large networks. More importantly, a comprehensive analysis of metabolic pathways can help identify critical biomarkers.

It is significant to note that in recent years, studies on gut microbes in animal models of hyperuricemia and gout have. Several studies have reported changes in the gut flora of people with hyperuricemia and found that some bacteria promote disease onset and progression (*Liu et al., 2020*). In animal models of hyperuricemia and gout, specific metabolites have been linked to intestinal flora metabolism (*Wang et al., 2020b*; *Song et*

*al., 2022*). A dysmetabolic intestinal flora may contribute to gout-related metabolism, thus promoting Th17 infiltration and further inflammatory symptoms. Several researchers have tried to address hyperuricemia through gut microbial mechanisms. The researchers searched for a probiotic strain (DM9218) capable of reducing UA levels and improving hyperuricemia (*Wang et al., 2019a*). In a study conducted by *Garcia-Arroyo et al. (2018)* uricolytic bacteria were shown to reduce SUA in hyperuricemia animals, suggesting possible prevention of hyperuricemia and renal damage. Although the specific mechanisms of gut microbes are unknown, they provide some constructive ideas for subsequent research on probiotic therapy for the disease. In addition, clinical studies on patients with hyperuricemia and gout have introduced diagnostic models based on gut microbial variability to facilitate differential diagnosis of the disease (*Guo et al., 2016*; *Yang et al., 2021b*). In particular, the diagnostic accuracy rate for gout patients is as high as 88.9% (*Guo et al., 2016*). Therefore, intestinal flora metabolism will be a new direction for future clinical research. The focus is on determining the mechanisms of gut microbial metabolism in patients with hyperuricemia and gout.

There is no doubt that the TCA cycle in question has a significant effect on hyperuricemia as well as on gout models, as illustrated above. A crucial aspect of the TCA cycle is that it forms an interconnected hub for other metabolic pathways, including the arginine, glycerophospholipid, pyrimidine, and tryptophan pathways, *etc.* There is speculation that there is another mechanism for hyperuricemia and gout that gut microbes could mediate. According to metabolic studies, hyperuricemia is associated with disorders of the primary pathway for bile acids or intestinal metabolism, which could provide a new direction for the treatment of this condition.

### Chinese Medicine and metabolomics

The available reference indicators cannot provide detailed insight into drug evaluation at present. More and more studies are exploring the mechanisms of traditional Chinese medicine (TCM) compounds for disease treatment and the efficacy of TCM through metabolomics (*Wang et al., 2017*; *Wu, Li & Zhang, 2019*). An experiment examined how Gout Party affected plasma metabolic profiles in an AGA rat model (*Wang et al., 2019b*). Compared to controls, the Gout Party treatment group affected 14 biomarkers, with the main pathways involved being fatty acid metabolism, bile acid metabolism, amino acid metabolism, and energy metabolism pathways. Not only that, but it also effectively reduced the release of inflammatory factors IL-6 and IL-8, exerting an anti-inflammatory effect. Another similar study investigated a mouse model of hyperuricemia by metabolomics. *Chen et al. (2021b)* presented that Tongfengxiaofang could regulate various disorders of arginine biosynthesis, galactose metabolism, pyrimidine metabolism, glycerophospholipid metabolism, tryptophan metabolism, and TCA cycle caused by hyperuricemia. The potential molecular mechanisms of anti-hyperuricemia in other herbs, such as QZTBD, EMW, and chicory, have also been reported (*Chen et al., 2016*; *Bian et al., 2018*; *Huang et al., 2019*). The current research in TCM focuses on animal studies of hyperuricemia and gout, and its overall characteristics are consistent with a comprehensive analysis of metabolomics. In addition, metabolomics can also be used to assess the performance of

a large class of Chinese medicine prescriptions besides studying the effects of drugs for disease treatment. In a study on high fructose combined with potassium oxonate-induced hyperuricemia, *Shan et al. (2021)* examined the effect of Ermiao wan categorized formulas (ECFs). ECFs included Ermiao wan (2 MW), Sanmiao wan (3 MW), and Simiao wan (4 MW). The results showed that two MW, three MW and four MW could partially modulate the disturbed lipid metabolic pathway. More importantly, 4 MW inhibited the disruption of TCA metabolism and purine metabolism induced by hyperuricemia better than 2 MW or 3 MW.

Lipidomics, an influential branch of metabolomics, has recently become a hot research topic and independent omics in biology. Technological advances enable lipidomics to identify individual lipid molecule species, and the integration of lipidomics with genomics and metabolomics can offer a holistic view of diseases (*Tabassum & Ripatti, 2021*; *Wang et al., 2020c*). An in-depth study of potassium oxonate-induced hyperuricemia was performed by *Tan et al. (2020)* utilizing metabolomics followed by transcriptomics. The plasma and urine metabolomics revealed 53 altered metabolites, 19 of which were lipids, primarily involved in the TCA cycle, lipid metabolism, amino acid metabolism, and pyrimidine metabolism. As a result of transcriptomics, genes that affect the cell cycle and energy metabolism have been identified. A key benefit of metabolomics is that it provides an understanding of the overall changes that occur in metabolism, making it excellent for assessing the therapeutic properties of herbal compounding with multi-targeted features. In other words, metabolomics allows for a deeper analysis of the organism's metabolites to identify the effective pathways through which drugs influence a disease.

Metabolomics has extensive experience in researching the pharmacological effects of herbal medicine used in the treatment of hyperuricemia and gout. TCM metabolomics studies primarily rely on animal models to evaluate its pharmacological effects. The first step in metabolomics studies of TCM is to dissect its possible active ingredients. A second step was to explore the overall metabolic changes induced by herbal medicine treatment in animals with hyperuricemia. Lastly, the metabolic network will be analyzed in order to elucidate TCM's molecular mechanisms of anti-hyperuricemia.

## CONCLUSIONS

Based on proteomic studies, we have identified a variety of potential "omics" biomarkers that provide etiological insights into the mechanisms of hyperuricemia and gout development. At the same time, the results of proteomic analyses suggest that some proteins involved in lipid metabolism, inflammation, and the complement system contribute to disease progression and may suggest research targets for disease treatment. In addition, metabolomics has been used to identify the important role of hyperuricemia and gout in metabolic disorders, and it is also an effective tool for Chinese medicine research. However, studies on diseases like hyperuricemia and gout have a relatively small sample size. Thus, whether the role of biomarkers is representative in diagnosing or suggesting disease remains to be investigated. Despite the fact that multi-omics data analysis is a great challenge, it turns out to be very rewarding at the same time. As part of future research in multi-omics

data analysis, reproducibility validation efforts of biomarkers will be an important focus of future research.

### Funding
This work was supported by the Science and Technology Plan Project of Gansu (21YF5FA126), and the Cuiying Scientific and Technological Innovation Program of Lanzhou University Second Hospital (CY2018-MS10). The funders had no role in study design, data collection and analysis, decision to publish, or preparation of the manuscript.

### Grant Disclosures
The following grant information was disclosed by the authors:
Science and Technology Plan Project of Gansu: 21YF5FA126.
Cuiying Scientific and Technological Innovation Program of Lanzhou University Second Hospital: CY2018-MS10.

### Competing Interests
The authors declare there are no competing interests.

### Author Contributions
- Xinghong Wu conceived and designed the experiments, performed the experiments, analyzed the data, prepared figures and/or tables, authored or reviewed drafts of the article, and approved the final draft.
- Chongge You conceived and designed the experiments, performed the experiments, authored or reviewed drafts of the article, and approved the final draft.

### Data Availability
This is a literature review.

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
