# Peer review of "The biomarkers discovery of hyperuricemia and gout: proteomics and metabolomics"

_PeerJ, doi:10.7717/peerj.14554_

## Round 0.1 · original submission · Major Revisions

The manuscript does not fulfill the standards established for the journal to be considered for publication in its current form. I agree with the reviewers that manuscript requires thorough revision and additional information to support the conclusion, significance and improve the quality of the publication. Please see the detailed comments made by the reviewers. Moreover, thorough English editing is required, as also suggested by both reviewers. Please contact a professional editing service to review your manuscript and revise the language to clear the context of the sentences at various places in the manuscript.

·

Basic reporting

This manuscript describes an important topic on clinical diagnosis or pathogenesis of hyperuricemia and gout. The topic has been ignored partially due to the advanced technologies available for diagnosis and the significant clinical manifestations of the disease. However, the structure of this MS seems too loose and the writing style is incohesive or poor fluency. This could be caused by the text being made by “machine translation” (from a non-English thesis) without manual editing. Therefore, the current form of the article needs to be edited by a professional scientific/medical editor to rephrase the whole text. It may be suggested that each section should have summery sentence (s) at the end.

Experimental design

Stated as above.

Validity of the findings

Specific comments:
1. In introduction: Line 62-93 too much method describing irrelevant to the topic or proportion to the whole section
2. Survey methodology should state the time duration of the literature research.
3. The mechanism of hyperuricemia and gout: missing hyperuricemia induced by uric acid production. The interrelationship of hyperuricemia and gout, eg. Hyperuricemia does not necessarily cause gout.
4. Inflammation caused by urate deposition: What about uric acid induced inflammation?
5. Proteomic study of hyperuricemia and gout: What are the differences between them?
6. Proteomics of other samples: What are the other samples? Urine, tissue, SF, MSU and kidney stone?
7. The “stones” of gout: the level of the subtitle needs to be reconsidered.
8. Line 468: TMAO and hippuric acid are “the” metabolites produced by bacteria?
9. Conclusion: too long to be eccentric.

Additional comments

the current form of the article needs to be edited by a professional scientific/medical editor to rephrase the whole text. It may be suggested that each section should have summery sentence (s) at the end.

·

Basic reporting

No comment

Experimental design

No comment

Validity of the findings

No comment

Additional comments

Line 54: The organism generates UA through purine metabolism. This sentence needs to be changed. This review is written in the context of human health.

Lines 56-58: The clinical manifestations of gout include asymptomatic hyperuricemia in the early stage, acute and chronic arthritis, and renal impairment. These statements are inaccurate. Please review the different stages of gout. Renal impairment is not always a manifestation of gout. Simply put, patients with gout have reduced kidney function or low eGFR.

132: Therefore, based on the urate transporter gene study, clinical research drugs to reduce UA have been developed (Sun et al., 133 2021). If this is the case, please provide more evidence and details about this drug. Otherwise, this is very confusing.

178: In contrast, regional and ethnic differences can lead to different protein expressions. There are no references to this statement. This is an important statement and needs more details.

Overall, this is a very long and dense review. There is an exceedingly high number of abbreviations that rendered the manuscript hard to read without having to look up the abbreviation. The method used for investigating the literature is weak and lacks the rigor of identifying the major article. In multiple sections, the article reads like a textbook. The authors could greatly benefit from having a rheumatologist review the manuscript. Also, a native English speaker reviewing the article is needed. The conclusion section is super long and doesn’t read like a conclusion but rather a summary page. Finally, I believe the authors need to focus on the type of literature to include in this review. Flipping between animals and humans is very confusing and weakened the clinical utility of the manuscript.

---

## Round 0.2 · accepted · Accept

Manuscript is significantly improved and authors have responded adequately to the reviewers comments. It can be accepted now in its current form.